# Comparative Analysis of Power Consumption between MQTT and HTTP Protocols in an IoT Platform Designed and Implemented for Remote Real-Time Monitoring of Long-Term Cold Chain Transport Operations

**DOI:** 10.3390/s23104896

**Published:** 2023-05-19

**Authors:** Heriberto J. Jara Ochoa, Raul Peña, Yoel Ledo Mezquita, Enrique Gonzalez, Sergio Camacho-Leon

**Affiliations:** Tecnologico de Monterrey, School of Engineering and Sciences, Ave. Eugenio Garza Sada 2501, Monterrey 64849, Mexico; a00813912@tec.mx (H.J.J.O.); raul.p.ortega@tec.mx (R.P.); yledo@tec.mx (Y.L.M.); msegonzalez@tec.mx (E.G.)

**Keywords:** Internet of Things (IoT), power consumption, Hypertext Transfer Protocol (HTTP), Message Queue Telemetry Transport (MQTT), long-term monitoring, NodeMCU

## Abstract

IoT platforms for the transportation industry are portable with limited battery life and need real-time and long-term monitoring operations. Since MQTT and HTTP are widely used as the main communication protocols in the IoT, it is imperative to analyze their power consumption to provide quantitative results that help maximize battery life in IoT transportation systems. Although is well known that MQTT consumes less power than HTTP, a comparative analysis of their power consumption with long-time tests and different conditions has not yet been conducted. In this sense, a design and validation of an electronic cost-efficient platform system for remote real-time monitoring is proposed using a NodeMCU module, in which experimentation is carried out for HTTP and MQTT with different QoS levels to make a comparison and demonstrate the differences in power consumption. Furthermore, we characterize the behavior of the batteries in the systems and compare the theoretical analysis with real long-time test results. The experimentation using the MQTT protocol with QoS 0 and 1 was successful, resulting in power savings of 6.03% and 8.33%, respectively, compared with HTTP, demonstrating many more hours in the duration of the batteries, which could be very useful in technological solutions for the transport industry.

## 1. Introduction

### 1.1. IoT Technology

In our modern world, considerable progress has been made in Internet of Things (IoT) technology. IoT systems have had exponential growth, and it is estimated that in future years, there will be more than 50 billion IoT devices in the world [1,2]. IoT technology has contributed to the development of novel logistics applications such as intelligent transportation systems [3], smart traffic monitoring [4,5], supply chain tracking [6], smart agriculture [7], damage detection, and monitoring the conditions in which goods are transported and stored [8], among many others.

According to the International Data Corporation (IDC), which is the premier global provider of market intelligence and advisory services, the transportation sector will have significant growth in the IoT market, and it is expected to reach USD 195 billion by 2020 [5]. IoT technology is changing the mode of operations in logistics and infrastructure, improving efficiency, reliability, safety, and tracking.

Even though at present, IoT platforms in the transportation industry are widely used, they have a big limitation with regard to their power consumption, as most of them are planned to be portable. Coupled with this, a change of battery is needed at certain intervals of time, particularly for long-term monitoring.

### 1.2. Long-Term Cold Chain Transport Operations

Effective cold chain transportation helps to reduce losses for companies. When a cold chain breaks down during the storage, transportation, and loading and unloading of products, quality and consumer health are compromised [9,10]. In the U.S.A., approximately 88 million tons of food are wasted year by year, representing almost USD 143 billion in losses [11]. Food spoilage due to the development of microorganisms is favored when important variations in environmental conditions such as temperature, humidity, and light intensity occur [12]. Moreover, products are often wasted when failures in refrigeration or transportation units take place. Conventional cold chain systems can only display and record environmental data in situ [13]; however, monitoring in real time is needed for tracking and predicting risks during cold chain transportation [14]. As aforementioned, it is clear that it is extremely important to find new methods and technologies to ensure the quality of products. Because of the great potential for monitoring abnormal events related to temperature and humidity and sensitive products in real time that facilitates taking corrective actions, IoT technology has rapidly been adopted in the supply chain area.

#### Technical Challenges in Cold Chain Transportation

Despite the benefits of IoT technology that enable interconnectivity between sensors, vehicles, and cloud-based IT systems, technical challenges need to be considered to maximize a system’s overall performance:Keeping devices connected as they move: A system’s architecture must be suited to managing bidirectional data transmission and intermittencies in network access when vehicles are in movement. Some IoT applications within cold chain transportation could require real-time data transmission over long distances.The implementation of embedded systems: Commonly, these kinds of devices have minimal resources, including low computation power. A lightweight and efficient protocol is required to optimize network bandwidth.Scalability: Broadcasting a message to or from many clients in real time is not an easy task. Additionally, it is hard to guarantee client–server connections when scaling up or down the number of devices connected to an IoT system.Reliable and secure data management: Identity, the encryption of messages, and authentication protocols must be used to protect data during the transmission process.Energy constraints: Even though at present, IoT platforms in the transportation industry are widely used, they have a big limitation with regard to power consumption, as most of them are planned to be portable. Coupled with this, a change of battery is needed at certain intervals of time, particularly for long-term monitoring.

### 1.3. Research Contribution

The aim of the presented work and the main contributions of this paper include the following:The design and implementation of a cost-efficient IoT electronic platform for the remote real-time monitoring of long-term cold chain transport operations.The evaluation and validation of the performance, accuracy, and precision of the sensed variables in the electronic platform.A comparative analysis of the power consumption between two of the most used protocols in the IoT, namely, the Hypertext Transfer Protocol (HTTP) and Message Queue Telemetry Transport (MQTT). The analysis was performed with identical hardware to compare the HTTP client/server and MQTT publish/subscribe models. Moreover, the magnitude of the quality of service (QoS) using the MQTT protocol was valued as either 0 or 1.A prediction model for the life cycle of batteries in the implemented platform, according to its specifications for long-term tests.

The rest of the paper is organized as follows. Section 2 describes the related work in terms of IoT architectures with NodeMCU and power consumption evaluations between the MQTT and HTTP protocols. Then, Section 3 discusses the main features of the MQTT and HTTP IoT protocols. Afterward, in Section 4, we describe the materials used to design the platform, and we depict the methods and setups to run the experiments. In Section 5, we describe the system prototype and the needed cost of each component. Then, Section 6 presents all the obtained results. This paper concludes in Section 7, wherein we provide our conclusions and future work.

## 2. Related Work

### 2.1. NodeMCU for IoT Applications

NodeMCU is a commonly used microcontroller unit (MCU) in wireless and IoT applications. It is a low-cost open-source IoT platform that includes firmware that runs in an ESP8266 Wi-Fi system on a chip (SoC) from Espressif Systems. It is a programmable, simple, and portable microcontroller that can be accessed and controlled remotely at any location across the globe at any time. NodeMCU can be used for many purposes such as domotics, smart plugs and lights, industrial wireless control, IP cameras, sensor networks, wearable electronics, and Wi-Fi location-aware devices, among many others [15]. Some remarkable experiments are mentioned below.

Khan et al. [16] used NodeMCU in a portable biometric attendance system for academic purposes with wireless interactions to a web server storing information in a MySQL database. All the data on a student are stored in the database when a finger is registered, and his/her attendance is counted automatically and wirelessly stored in the designed database.

Wi-Fi microcontrollers are also frequently used in domotics. They are not only used for status indicators of different variables but also for efficient control and optimization performance, avoiding the unnecessary use of power and resources by turning on/off lights, regulating their intensity, and controlling fan and water pump flow [17,18,19].

Abdulahad [20] elaborated a system to control air quality, temperature, and humidity in many food stores in remote locations through web servers. Two actions are taken: an air cooler is turned on to cool down perishable food when some variables reach certain values, and an air puller is activated to pull the contamination out of the locations.

Furthermore, health is an important area in which improvements in IoT technology can always be made. Chooruang [21] implemented a heart rate monitoring system using an ESP8266 Wi-Fi module (AMICA, Taiwan) in which detection of the real-time heart rate of a patient was realized, obtaining a very good percentage error of approximately 2% and 6%. Besides heart rate monitoring, blood pressure monitoring experiments have been conducted. Singh [22] created an economically user-friendly method to detect blood pressure in real time. The prediction program can automatically release an appropriate medical dosage invasively in cases of an emergency. In addition, blood pressure readings are immediately communicated to emergency contacts if the data are outside of the threshold.

Some other investigations and experiments have been conducted in the IoT area that will have a lot of applications in the near future such as in voice-controlled autonomous vehicles [23]. IoT applications are limitless and can be used in any area and in any environment with the correct sensors, actuators, and displays.

### 2.2. Power Consumption between HTTP and MQTT

In terms of the application layer, which is responsible for the interface between communications and the application running on the host [24], two of the most representative application protocols for IoT technology, namely, HTTP and MQTT, are compared. These two protocols play important roles because both protocols are used in the experimentation part in Section 5, which gives us important results in Section 6 related to the power consumption of the system.

Wireless communication protocols represent a percentage of the total power consumption in an IoT platform, in which the battery depends on the selected protocol. Below, some experiments evaluating power consumption using the HTTP and MQTT protocols are presented.

In [25], energy consumption studies on 3G and Wi-Fi transmission using real traffic data were conducted. The studies were performed with a modern smartphone for Wi-Fi, a specific mobile broadband module for 3G, and a major mobile operator, where the application was run using the MQTT and HTTP protocols. MQTT resulted as a better solution for energy consumption when the number of users and the sharing interval were low.

An air control device is presented in [26]. It uses an ESP12E microcontroller and a DHT22 temperature sensor in a system using Wi-Fi connectivity. Different parameters such as latency and current consumption are compared to obtain the most optimal case. The paper concludes that the MQTT protocol has lower latency and overhead and power transmission than HTTP, and the battery life of the batteries becomes longer with MQTT.

In [27], an evaluation of many established messaging protocols including MQTT, the Constrained Application Protocol (CoAP), and HTTP for IoT systems is conducted, comparing architectures, transport protocols, message size, message overload, latency, bandwidth, reliability of QoS, and interoperability, among other variables. It establishes that the MQTT protocol consumes less power than HTTP and that it is better for power consumption solutions.

According to [28], MQTT resulted in a better power consumption performance. In the experimentation, with 3G technology, 4.1% of the battery per day is saved just by using MQTT over HTTPS to maintain an open stable connection, while Wi-Fi technology had similar results as well. Finally, in [29], the authors compare MQTT with QoS 0 vs. HTTPS protocols, evaluating their performance and battery energy consumption with a novel approach in which a developed device acts as an MQTT broker instead of the typical cloud-based architecture, eliminating the need for an external internet server and making the system simpler and more affordable and secure. It concludes that HTTPS is slightly more efficient in terms of establishing connections, while MQTT is more efficient during transmission and regarding power consumption.

Most of the previously presented works agree that the MQTT protocol consumes less power than HTTP, which can be used as a very good solution in IoT technology for power consumption issues.

## 3. IoT Application Layer Protocols

### 3.1. HTTP

HTTP is the first acknowledged IoT protocol that uses a request/response architecture in a client–server model, and it is mainly used to deploy web servers. A web server provides solicited data via browsers, and through HTTP, they are delivered in HyperText Markup Language (HTML) format. The browsers always begin communication with the petition of an HTML document to the server, and then, the document is processed and sends more petitions to request scripts, Cascading Style Sheets (CSS), among other aspects.

#### Client/Server Architecture

The client/server architecture is a request–response model consisting of a client and a server system communicating over a computer network. This architecture provides an enhanced way to share the workload. The client is constantly launching a connection to the server, while the server is always waiting for a request from any client. Giving an example, as seen in Figure 1, a client can be a computer hardware device and a server can be a computer. Each of the servers provides a response to the client devices (laptops, tablets, and smartphones) [30,31].

### 3.2. MQTT Protocol

MQTT, developed in 1999, is an open network protocol that transports messages between devices and is frequently used in IoT applications. The Transmission Control Protocol (TCP) is one of the fundamental sets of protocols on the internet, which is important in the MQTT protocol because it usually runs over the TCP/IP.

Unlike HTTP, the MQTT protocol uses a publish/subscribe architecture, and, although HTTP is more used, MQTT has more applications in the wireless area because it is faster, requires less bandwidth, and requires less consuming power. An example of an MQTT application in the wireless area is in the Web of Things platform (WoT) for students’ learning process [32]. The WoT performs remote experimentation with a collaborative learning environment and analyzes the data generated via sensors displaying relevant information on dashboards. The authors define three layers for a full cycle of development in IoT solutions: (1) basic interaction with sensors and specific communication protocols; (2) data management models to handle the generated data; and (3) the processing and visualization of the most relevant indicators in the IoT devices.

One of the main advantages of the MQTT protocol is the energy saving of the broker when a cloud-based broker is selected because, in many experiments, a physical broker must be used to transmit the information, such as a Raspberry Pi computer. Furthermore, the power savings are because it takes less time, and fewer data packets (bigger data-size packets than HTML) are sent.

#### 3.2.1. Publish/Subscribe Architecture

In the publish/subscribe architecture, as seen in Figure 2, a client publishes information and other clients can subscribe to only the information they want to receive. Publishing is the operation in which a client wants to send data to the broker and subscribes when the broker sends the data to the clients. While HTTP requests to open and close the connection at each request, MQTT stays online to always maintain an open channel between the broker and the clients. Some of the challenges in this architecture are the discovery of publishers and topics and the guaranteed delivery when publishing to a distributed database [33].

Regarding the structure of this architecture, MQTT topics are structured in a hierarchical way, similar to folders in a computer where a slash (/) establishes the limit. One example of this topical layer structure could be myhome/upperfloor/myroom/temperature.

Regardless of the QoS level, the interaction sequence in MQTT would be the connection between the publisher and the broker where an acknowledgment signal is returned, the connection and subscription between the subscriber and the broker with its acknowledgment signal, and, finally, the publishing from the publisher to the broker and then to the subscriber if it is a subscriber to that topic.

#### 3.2.2. Broker

The MQTT broker is the heart of the entire MQTT protocol because it is the server that transfers messages between senders and receivers. The broker facilitates and filters the information between clients. This has many advantages in a system [34]:Space: Subscribers and clients do not need to know the IPs of each other.Time: Clients do not have to be running at the same time.Synchronization: Publishing and receiving information can occur simultaneously, and this does not halt operations.

#### 3.2.3. Quality of Service

All messages are published with a QoS level, which specifies the delivery requirement and ensures the reliability of messaging. MQTT supports three levels of QoS: 0, 1, and 2 [35,36], which can be seen in Figure 3. In a system, clients can have different numbers of QoS levels, and when a client is subscribed to a specific topic, the client determines the maximum level of QoS. It does not matter if the sender publishes a message on a certain topic with a higher QoS level; the system will proceed with the level established by the receiver [37].

QoS 0: “At most once”. A message is sent only once, and whether the message was received by the client is not verified. This level is the simplest, but it is possible to lose packages of data.QoS 1: “At least once”. At level 1, a message is sent, and the delivery status is checked via the status check message, called the PUBACK. The broker stores messages and sends them until clients have acknowledged their delivery, but if a PUBACK is lost, duplicate messages are received by the client since there is no confirmation of delivery.QoS 2: “Exactly once”. Messages have a second acknowledgment round trip, having four interactions between the client and the broker to ensure that the messages are delivered only once. Message loss is not possible with QoS 2, but the process is more complicated and sometimes requires more time.

Comparing the different QoS levels, in [38], cellular network and Wi-Fi experimentation was conducted to make a comparison in terms of power consumption using the MQTT protocol. The experimentation concluded that QoS 1 was the level that consumed less power, followed by QoS 2 and finally QoS 0, over Transport Layer Security (TLS). QoS 0 and 2 consumed 6.7% and 5% more energy than QoS 1, respectively.

In [39], a power consumption comparative analysis using MQTT was made with different variables in the system, such as the number of publishers and subscribers, different amounts of payloads, and different magnitudes of QoS levels. In general, QoS 1 and 2 had better power consumption results compared with QoS 0. Moreover, as the QoS level increased, the number of messages reaching the subscriber in a given period was reduced.

## 4. Material and Methods

### 4.1. Hardware

In the proposed electronic platform, different devices were used for our experimental work: a NodeMCU board for data capture, processing, and transmission; a DHT11 module to collect temperature and humidity measurements; a GY-NEO6MV2 module, which is a Global Positioning System (GPS) sensor for geolocation; and a microSD card module that allowed the NodeMCU board to communicate with the memory card and write or read the information on it. To visualize the relevant data, a Grove liquid-crystal display (LCD) with an RGB (red, green, and blue) backlight was included. Finally, a portable battery took over to power the whole system platform. The different components can be seen in Figure 4.

### 4.2. Measurement Equipment

To analyze the behavior of the system, the oscilloscope Tektronix TBS1000 (Oregon, USA), a MUL-280 multimeter (Longwood, FL, USA), and an NI myDAQ (Austin, TX, USA) data acquirer were needed to measure accurate values of voltage, current, power, and energy wasted on the electronic platform. Figure 5 shows the mentioned measurement equipment.

#### Data Acquirer

Data acquisition (DAQ) in electronic prototypes is the process of measuring an electric signal, such as voltage or current, with a computer in programmable software. Transductors are essential to calculate physical phenomena such as temperature, pressure, light, sound, force, and acceleration, among many others, through measurements of changes in voltage and current. In comparison with traditional measurement systems such as oscilloscopes and multimeters, the DAQ system takes advantage of processing power, productivity, visualization, and connectivity skills in the industry with powerful, flexible, and cost-effective measurement solutions.

A DAQ device is an interface between a computer and physical signals. It mainly digitizes analog input signals to be interpreted with a computer. Three key components are needed in a DAQ device to measure a signal:Signal conditioning: Physical signals are usually noisy, and they are not prepared for input into an ADC. Signal conditioning is a process that adapts signals to a required input range. This circuit can include amplification, attenuation, and filtering.Analog-to-digital converter (ADC): Analog signals from sensors are converted into digital signals before being sent to the computer. The communication between the DAQ system and the computer is digital, so the DAQ system carries out the procedure to transform analog signals into digital signals.PC bus: The communication between the DAQ and the PC is through a port where the bus works as a communication interface. The ADC makes periodic samples of the signal at a predefined rate, which are sent to the PC through the PC bus, where the original signal is reconstructed. The most common PC buses are USB and Ethernet.

A NI myDAQ, as shown in Figure 5c, was needed to record readings in the system. It can be used as an oscilloscope, ammeter, voltmeter, or function generator, among many other applications. The NI myDAQ uses LabVIEW and NI Multisim software and information is sent via a USB PC bus. It has a 16-bit resolution and a maximum sampling rate of 200 KS/s. It can work with ± 10V and has a very good time resolution of 10 ns.

### 4.3. Web Development Methodology (Front-End)

In this prototype, the creation of a web server was very important to display the sensed data acquired with the microcontroller. HTML, CSS, and JavaScript were some tools required for the elaboration of the front end.

#### 4.3.1. HTTP Dashboard

Figure 6a illustrates the dashboard design. First, the indicators of the variables are displayed showing the actual magnitudes of the temperature and humidity and the coordinates, and they are updated every 10 s. The update time can be changed to any time greater than 2 s, which is the period in which the DHT11 sensor can work with more frequency, but for this prototype, it was decided to make lectures every 10 s to have enough space in the memory. Then, a map in real time is displayed, as can be seen in Figure 6b. A map design from “mapbox” was obtained, which is an American provider of online maps for websites such as Facebook, Snapchat, and The Weather Channel, among many others. It has a free option and is a little bit limited, but for this project’s purposes, it worked perfectly. Finally, Figure 6c illustrates 2 graphs of the last 30 registers of the temperature and humidity measurements, which are updated every 10 s as the indicators at the top of the web server. When the cursor is on one point of the graph, it displays the date and the magnitude.

#### 4.3.2. MQTT Dashboard

One of the advantages of using MQTT is that programming in Arduino IDE is easier than in HTML because it is not necessary to make the design. With the help of an MQTT broker, only a connection between the microcontroller and the broker is required. In this case, an Adafruit broker was used, and the design of the dashboard was directly made on the web page. It is only necessary to give the correct variables to display them in different formats. As shown in Figure 7a, the MQTT dashboard was made, and it indicates the temperature and humidity of the system. One disadvantage is that the broker is limited to many dashboards during free trials.

For geolocation purposes, a map in real time is displayed. The cloud-based MQTT Adafruit IO broker is limited to only one map designer, and in this case, this is the company Leaflet. Adafruit makes everything suitable for its products, and it designed libraries focused on good communication between the physical devices and the broker and map. Communication between a selected GPS sensor and the interaction with the map has its challenges because many tools are required to guarantee this. Finally, with the help of these tools, the broker was able to display our location and our trajectory in predefined periods. In Section 6, many periods of time are defined to analyze the waste of power in function of the number of messages sent with the different protocols. In Figure 7b, the route that was taken to travel along with the current location can be seen. This location is displayed and updated every predefined time according to the experimentation.

### 4.4. Web Development Methodology (Back-End)

The back end is in charge of the functionality of the web page in the server-side development. It is important to carry out some activities that are not easily seen, such as communication with the server, the application, and the database. This is what happens when performing any action on a website. For our project, Hypertext Preprocessor (PHP) and MySQL tools were used in the back-end web development.

#### 4.4.1. Database

MySQL is an open-source database management service that uses Structured Query Language (SQL) as a specific domain language that is widely used in programming. It is the most popular system used with PHP, and it is very fast, reliable, and easy to use. A block diagram of the database is presented in Figure 8, where there are wired connections between the sensors and the NodeMCU. The microcontroller connects via Wi-Fi to the local access point (router) via HTTP and connects to the domain and the hosting server. With the code “post-nodemcu.php”, it uploads the information to the MySQL database, and with “nodemcu.php”, the information can be visualized from anywhere with the correct Uniform Resource Locator (URL).

#### 4.4.2. PHP

PHP is a server-side scripting language that is especially suited to web development. PHP is used to connect and manipulate databases. Two codes, “nodemcu.php” and “post-nodemcu.php”, were created to prepare the database to store the readings.

“nodemcu.php”: This code was created to allow data visualization everywhere. With this code, a web server in the created domain is made to have the readings sent to the database only by accessing the right link.“post-nodemcu.php”: This code is in charge of publishing sensor readings to the MySQL database.

The PHP code creates an index of the table and prepares it to store the necessary readings in the MySQL database. In any system, it is important to save data when the electronic platform is working for future analysis if required.

In Figure 9, the interface in phpMyAdmin can be seen, where all the values of the temperature, humidity, latitude, and longitude readings are stored in conjunction with their identification numbers and the times of the readings. Each reading was performed every 10 s.

### 4.5. Measurement Methodology

Many tools and devices were used to facilitate the different measurements in the circuit. A multimeter and an oscilloscope were used to see, in the first instance, the operation range of the current and the voltage consumed in the electronic platform.

The determination of the instantaneous power of the electronic platform to conduct statistical procedures and obtain conclusions about the system was significant. With the multimeter and oscilloscope, it was not possible to carry out those readings; moreover, it was noted that there was too much variation in the current in the system according to the readings with the multimeter.

In that sense, it was necessary to elaborate a program to store the values of instantaneous power
(1)$$P_{\mathrm{instantaneous}}\left(n\right)=V(n)\;\times \;I(n)$$
and obtain the values of the voltage and current at the same time. At the end of the experiment, a buffer of lots of readings of the instantaneous power makes the statistical analysis more robust than with oscilloscope and multimeter readings.

For the above-mentioned purpose, a small program in LabVIEW 2019 software was developed, wherein the data acquirer NI myDAQ was used to obtain voltage and current readings to determine the power of our circuit using the different protocols and with the different variables determined during this project. The objective of the LabVIEW program is to capture instant power readings, which are exported to a spreadsheet in Microsoft Excel to proceed with the statistical procedures.

A DAQ assistant, which is a based tool that makes the steps to acquiring simple measurements in the LabVIEW configuration easier, was needed. Two DAQ assistants were necessary in the programming to perform the readings: one for voltage and the other one for current. Analog inputs (AI) were used for the voltage readings and multimeter inputs (A) for the current readings. The location of the multimeter inputs is on one side of the analog and digital pins of the device, as can be seen in Figure 10a,b.

When setting up each DAQ assistant, specifying the signals that are going to be acquired and declaring them as analog inputs is needed. Voltage readings are assigned to “ai0” (analog inputs), while current readings are assigned to “dmm” (digital multimeter) pins. Once the configuration is carried out, the software shows the correct way to perform the connections.

The current measurement diagram can be seen in Figure 10a, while the voltage measurement diagram is shown in Figure 10b. As was mentioned, in the current measurement, A0, AI, and DIO were not used, and only the HI and COM ports in the NI myDAQ were used as an ammeter. Moreover, in the voltage measurement, the HI and COM ports were not used, and only the AI0 (analog inputs) in the NI myDAQ were used as a signal recorder.

Figure 10c depicts the complete schematic of the proposed electronic platform with the voltage and current measurements using the NI myDAQ. The readings are sent via USB to the computer to record all the data. Once the readings are saved, statistical procedures can proceed to carry out the comparative analysis.

The main objective of the LabVIEW program is to store lots of instant power data on our electronic platform. In Figure 11a, the interface on the front panel of the LabVIEW program can be seen, wherein three indicators and three waveform charts can be appreciated. In the charts, the readings are recorded according to the programming in the block diagram and then exported to a Microsoft Excel spreadsheet.

In Figure 11b, the block diagram programming in LabVIEW can be seen. It is a very simple program, which consists of one for-loop that oversees carrying out a reading in each iteration. The DAQ assistant is in charge of the current readings and DAQ assistant2 oversees those of the voltage. These two signals are multiplied to obtain the power of the system, and then it is displayed in a chart, where a total of 100 (N = 100) current, voltage, and power readings are recorded. The time between each iteration is 1.5 s, so for each test, 150 s (2.5 min) is required.

## 5. System Prototype

In this section, the architecture and price of the proposed electronic prototype are presented. A block diagram of the portable electric prototype can be seen in Figure 12a. The different serial protocols that communicate with the sensors, storage, and display can be seen at the bottom of the diagram, where UART is used for the GPS, SPI for the microSD, and I2C for the LCD. Then, the prototype establishes communication via HTTP or MQTT with the web server or with the Adafruit cloud-based broker, depending on which experimentation analysis is required. Communication with the MySQL database is realized in both cases.

The cloud-based broker Adafruit IO is used in this project. Although the free version has many limitations, such as limited feeds variables (10), limited dashboards (5), and a limited sampling rate (30 data/minute), it perfectly meets our prototype specifications. Then, communication between the microcontroller and the final clients through wireless networks is realized to display the sensor readings on the dashboards in real time. Finally, the physical prototype can be observed in Figure 12b, where all the devices used in this electronic platform can be seen.

### Price

The cost of the system is usually considered one of the three primary constraints of any project (scope and time are the other two). It is important and depends on us making the electronic platform as inexpensive as possible without limiting functionalities. In Table 1, the components of our electric platform with their prices can be seen to make an estimate of the budget required to elaborate this prototype. The prices are seen in dollars and are according to the Alibaba company website.

With USD 22.61, the electric platform can be put together. Many times, in IoT projects, LCDs are not required because the information can be seen on the dashboards or in the cloud. Therefore, with only USD 13.52, a complete real-time system with many sensors can be installed.

## 6. Results

In this section, a comparative analysis of the power consumption between the HTTP and MQTT protocols is conducted. In addition, the prediction model for the life cycle of batteries is presented.

### 6.1. Power Consumption of HTTP and MQTT with QoS 0 and 1

A comparison of the consumed power between the HTTP and MQTT protocols is presented in Table 2. A total of 3 tests for each of the 100 samples were carried out for 2 different situations for HTTP and MQTT with QoS 0 and 1. For each protocol, readings of the electronic platform were performed, one with the complete working system and another connecting only the microcontroller to analyze the power wasted in the NodeMCU without sensors, displays, or memory. In this table, the current, voltage, power, and energy were analyzed. In each case, the minimum and maximum current, voltage, and instant power were obtained, as well as the average power to compare them. From this data, the energy wasted in each test was calculated by multiplying the power average times the time for which the tests lasted (150 s).

In addition, the averages of the previous results were obtained. First, the amounts of power wasted in the circuits in which only the microcontroller was working (without sensors, displays, or memory) were estimated to be very similar, independent of the protocol. Only 3.24 mW of power was the biggest difference between the average powers in the 3 cases. HTTP wasted 381.62 mW, while with the MQTT protocol, 380.17 mW was wasted with QoS 0 and 383.31 mW with QoS 1. It can be declared that there are no significant differences between these cases when energizing only the microcontroller.

Secondly, a difference between the average powers can be noted. Theoretically, the MQTT protocol with any QoS level wastes less power in its operation than HTTP. In the experimentation, a successful result was obtained. With MQTT, the circuit was consuming on average 629.68 mW with QoS 0 and 614.33 mW with QoS 1, while with HTTP the average power was 667.33 mW.

The magnitude of the difference in the average power consumption between the protocol that consumed less power (MQTT with QoS 1) and the protocol that consumed more power (HTTP) was 55.83 mW. The difference between HTTP and the MQTT protocol with QoS 0 was 40.48 mW. Finally, the least difference in power consumption was with the MQTT protocol between QoS 0 and 1, being 15.35 mW.

In Figure 13, the information in Table 2 is illustrated as a graph. The HTTP readings are shifted up with respect to the two cases in the MQTT protocol, meaning that HTTP consumes more power than the MQTT protocols. The downward peaks for the MQTT protocols can be observed in a better way, where QoS 1 has more than QoS 0.

The relationship between the above-mentioned cases in terms of power rate is presented in the following equations:(2)$$\mathrm{Power}\;\mathrm{rate}=\frac{P_{\mathrm{MQTT}0}}{P_{\mathrm{HTTP}}}=\frac{629.68\;\mathrm{mW}}{670.16\;\mathrm{mW}}=0.9396$$
(3)$$\mathrm{Power}\;\mathrm{rate}=\frac{P_{\mathrm{MQTT}1}}{P_{\mathrm{HTTP}}}=\frac{614.33\;\mathrm{mW}}{670.16\;\mathrm{mW}}=0.9167$$
(4)$$\mathrm{Power}\;\mathrm{rate}=\frac{P_{\mathrm{MQTT}1}}{P_{\mathrm{MQTT}0}}=\frac{614.33\;\mathrm{mW}}{629.68\;\mathrm{mW}}=0.9756$$
which obtain the relation percentages of power consumption in the different protocols.

Different power rates of 0.9396, 0.9167, and 0.9756 are represented, meaning that the MQTT protocol with QoS 1 consumes 91.67% and 97.56% of the total average power consumed in HTTP and MQTT with QoS 0, respectively, and the MQTT protocol with QoS 0 consumed 93.67% of the power consumed in HTTP. In other words, there are power consumption savings in the MQTT cases of 6.03% (QoS 0) and 8.33% (QoS 1) with respect to HTTP.

With the previous data, the expected lifetime of any power supply can be calculated. In the next equation,
(5)$$\mathrm{Battery}\;\mathrm{Life}\;(h)=\frac{\mathrm{Battery}\;\mathrm{Capacity}\;(\mathrm{Wh})}{\mathrm{Estimated}\;\mathrm{Power}\;(W)}$$
the battery life in hours can be calculated for the different protocols. The battery capacity (Wh) was 72 Wh according to the battery specifications, while the average power was taken as the estimated power.

Only one calculation was carried out from the data where only the microcontrollers were energized because the average powers were very similar. The average of the three values was taken to determine the battery life, while in the complete system, the data were taken from the average powers viewed in Table 2.

Furthermore, the rate of change in the autonomy associated with the useful time of the battery can be seen in Table 3. This can be seen as the percentage over time, where 100% is divided over the calculation of the battery life to obtain the rate of change (%/h).

With HTTP, the battery life is expected to last 107 h, while in the MQTT protocol with QoS 0 and 1, it is expected to last 114 and 117 h, respectively. The expected time savings with the MQTT protocol for this battery are 7 and 10 h. As a reference, when only energizing the microcontroller, the system is expected to last almost 189 h in all cases.

### 6.2. Normal and Real Distribution of Instantaneous Power Consumption

In Table 4, the 3 tests per protocol are put together, forming a 300 instantaneous power sample table for each one of the cases: HTTP, MQTT QoS 0, and MQTT QoS 1.

The prediction model will be presented in Section 6.3, and it is based on the statistical normalization of 300 instantaneous power samples, i.e., their distributions are aligned to a normal (Gaussian) distribution, as seen in Figure 14a. In this sense, Figure 14a shows the following main statistical parameters: the minimum and maximum values, average, standard deviation, variance, median, mode, and range between the maximum and minimum values.

In Figure 14b, the real distributions of power consumption can be observed, comparing the three different protocols and analyzing the most constant range of values presented in the experimentation. The HTTP curve is shifted right, meaning that is the one that consumes more power, while MQTT QoS 1 is the protocol that consumes less power.

### 6.3. Prediction Model for Long-Term Tests

The previous results can be used to characterize the behavior of the batteries in the systems with different protocols. Long-term experiments were made to corroborate and analyze this characterization. The results of the duration of the battery and the relative errors are analyzed in Table 5.

In addition, the model assumes a linear discharge of the battery, which is validated by the experimental data collected over 84 h and plotted in Figure 15. A steeper slope is noted for HTTP with respect to the two-line graphs for the MQTT protocol. This graph shows that the battery in the MQTT protocol with QoS 1 will last more than the others because it consumes less energy. With the line graphs, the theoretical values are presented, while the points indicate the real measurement values.

The linear equations of the battery capacity are given via the next equation:(6)%Battery=100 − mt
where ‘m’ is equal to the slope, and ‘t’ is the time in hours. The number 100 is the maximum percentage of the battery that will decrease in time given the rate of change ‘m’.

Different values of the rate of change are seen in the three different cases of the HTTP and MQTT protocols. Next, the theoretical linear equations
(7)%Battery MQTT 1=100 − 0.8532t
(8)%Battery MQTT 0=100 − 0.8746t
(9)%Battery HTTP=100 − 0.9308t
are presented.

With the results of the long-term experimentation, the real linear equations
(10)%Battery MQTT 1=101.25 − 0.9077t
(11)%Battery MQTT 0=101.17 − 0.9236t
(12)%Battery HTTP=100.58 − 1.002t
are obtained, which are very similar to the theoretical linear equations previously seen.

In Table 6, the theoretical energy consumption data from the longtime tests can be seen, while Figure 16 contains the theoretical graphs.

#### Relative Error between Theory and Experimentation

The relative error is given using the next equation:(13)$${\mathrm{Error}}_{p}(n)=\frac{\left|\mathrm{Theoretical}\;\mathrm{value}-\mathrm{Experimentation}\;\mathrm{value}\right|}{\mathrm{Theoretical}\;\mathrm{value}}\times 100$$

In Figure 17, the relative error percentage graph of the three cases is given. An increase in the error percentage is noted because the error accumulates as time passes. The relative error for HTTP is higher than for the other two cases of the MQTT protocol. Both cases behave in a very similar way.

Beyond 60 h, the relative error does not behave in a linear way, and it behaves as a higher-order function, as can be appreciated in the graph.

## 7. Conclusions and Future Work

In this paper, we presented the design and implementation of a cost-efficient IoT electronic platform for the remote real-time monitoring of long-term cold chain transport operations. In terms of cost, with USD 22.61, the electric platform can be put together. Moreover, if an LCD is not required in the implementation, with only USD 13.52, a complete real-time system can be installed. In terms of data validation, for both the HTTP and MQTT implementations, a dashboard and a map with real-time data were presented. The displayed data show the actual magnitude of the temperature and humidity and the coordinates for geolocation.

Using our prototype, we conducted experiments to compare the HTTP and MQTT protocols with different QoS levels to select better solutions in the IoT area for the transport industry. In terms of the MQTT protocol, QoS 0 and QoS 1 were selected as the extreme cases in power consumption between the following three levels: QoS 0, QoS 1, and QoS 2. According to [38], QoS 1 is the level that consumes less power and QoS 0 is that which consumes the most. Nevertheless, for future experimentation, we will integrate and validate QoS 2 in our test platform.

One of the main advantages of the MQTT protocol over HTTP in the IoT is that it consumes less power, which is a very important variable in IoT technology, and it could be proved in the experimentation. The above seems to be related to the size of the payload header format and the fact that, unlike MQTT, HTTP treats one request at a time [40].

In our results, lots of data on instantaneous power were obtained with an NI myDAQ as the voltage and current data acquirer. The MQTT and HTTP protocols with a sampling rate of 10 s between each transfer of data were analyzed and 15 min between the data transfer to the MySQL database. The entire electronic platform consumed on average 629.68 mW and 614.33 mW using the MQTT protocol with QoS 0 and 1, respectively, while when using HTTP, the circuit consumed 667.33 mW. The electronic platform using the MQTT protocol with QoS 1 resulted in 91.67% of the total average consumed power in the electronic platform with HTTP, and 97.56% of the power wasted with QoS 0, while QoS 0 resulted in 93.96% of the power wasted in HTTP. There were power consumption savings in the MQTT cases of 6.03% (QoS 0) and 8.33% (QoS 1) with respect to HTTP. Using the MQTT protocol with QoS 1, the battery is expected to last 10 h more than when using HTTP and 3 h more than with QoS 0 with a 72 Wh portable battery. With QoS 0, the battery is expected to last 7 h more than with HTTP. In cargo transport, 7 and 10 more operating hours result in a very good improvement in the system, having variables readings working correctly over more time. In the experimentation section, HTTP resulted in a bigger discharge rate than the MQTT protocols.

Another advantage of the MQTT protocol is that no energy must be wasted in the broker if using a cloud-based broker, as was the one used in the presented prototype. A device-to-gateway model was used in the MQTT protocol where the Adafruit IO cloud-based broker acted as a cloud-based gateway. In other cases, it is necessary to use another microcontroller as a broker to transmit information.

Furthermore, we presented a prediction model for the life cycle of batteries, according to their specifications for long-term tests. The obtained results show that the characterization for a few hours (from 0 to 48 h) only presents a small relative error of up to 2.24 in comparison with the real experimentation. As time passes, an increase in the error percentage is observed because the error accumulates. The relative error in HTTP is higher than in the other two cases of the MQTT protocol. Both cases behave in a very similar way.

A limitation of this article is that no analysis of how fast any system can work was carried out. It is important to have in mind the type of application in which your platform is going to perform. However, in this project, fast readings are not necessary because the main idea is to give contributions to future works to reduce the power of the circuit. In the IoT, it is very important to take care of wasted energy because most of the sensors in a sensor network are connected to a small portable battery, which has a limited lifetime. In the HTTP experimentation, it was realized that information can be transmitted very fast, but also, the system is going to be limited to the specifications of all the components of the circuit. In this case, a DHT11 temperature and humidity sensor can read a maximum of data every two seconds. In many cases, the velocity of a system depends on the physical limitations of the sensors. Another relevant topic to discuss, which was not included in the present work, is the evaluation of the differences in power consumption in a system with the ability to switch between the different qualities of service offered by the MQTT protocol.

According to all the presented results, we depict some potential strategies to optimize power consumption in IoT-enabled cold chain systems:Develop new strategies for the detection of abnormalities in the parameters being monitored. New methodologies and systems need to consider the criteria to define abnormalities and detect whether transportation vehicles are in movement or not due to environmental conditions that could impact the desired range of values for temperature and humidity in products.Set the sampling rate according to the needs of the application, taking into consideration that the sampling frequency is proportional to the power consumption of any system. It would be valuable to increase the sampling period without altering the main functions of the electronic platform.Develop a hybrid IoT system capable of switching between the different levels of services in the MQTT message delivery quality of service according to the type of product, the environmental conditions, and the traffic within the cold supply chain.Leave out devices that are not indispensable to the performance of the system such as LCD, which is used for debugging purposes, and microSD memory, which can be substituted with the cloud database.Apply one of the multiple specialized methods and techniques to optimize code, which is intended to become smaller, executing tasks in a quicker way and performing fewer operations. With code optimization, the program is expected to consume less memory and the electronic system less power.

Finally, this investigation opens the discussion for continuing this work in the future in the following different areas:Implementing cellular networks and analyzing the consumed power in the electronic platform.Fabricating the electronic platform in a PCB-Microelectro-Mechanical System (MEMS) to significantly reduce its size and make it more portable.Experimenting with different sampling rates to determine differences in power.Associating the data storage and databases with cloud providers such as Alibaba Cloud and Amazon Web Services, among others.Studying the differences in power consumption when varying many variables such as the time of data transmission, the number of subscribers/publishers in the system, and the number of sensors in wireless network nodes.Exploring other application layer protocols such as CoAP and WebSocket, which also have many uses in actual IoT applications.

## Figures and Tables

**Figure 1 sensors-23-04896-f001:**
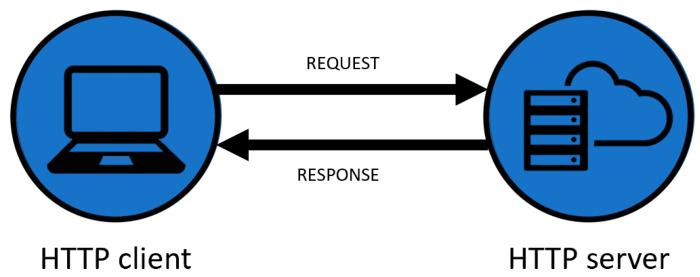
Request–response architecture in HTTP.

**Figure 2 sensors-23-04896-f002:**
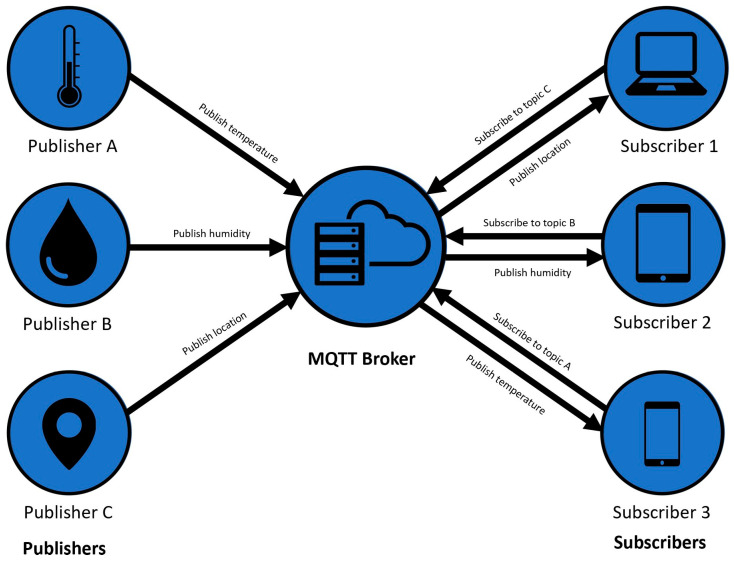
Publish/subscribe architecture in MQTT protocol.

**Figure 3 sensors-23-04896-f003:**
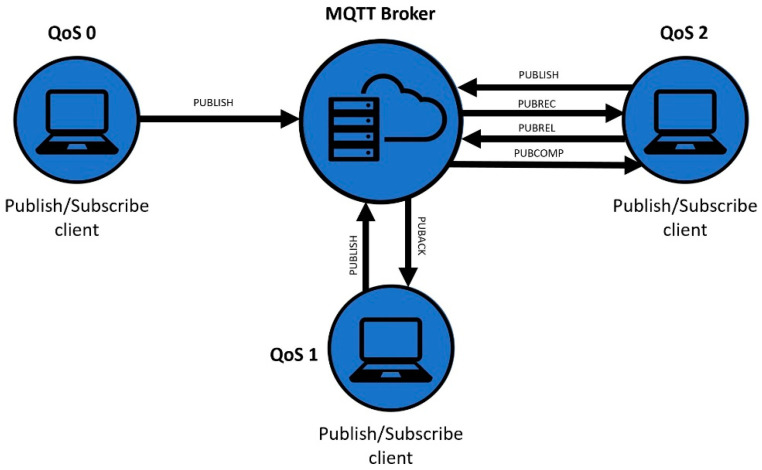
Packet transmission method with QoS levels in MQTT.

**Figure 4 sensors-23-04896-f004:**
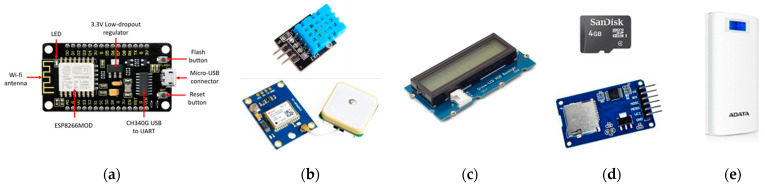
Devices used to design the experimental electronic platform: (**a**) NodeMCU board; (**b**) DHT11 module and GPS sensor; (**c**) LCD; (**d**) microSD card module and memory; and (**e**) portable battery.

**Figure 5 sensors-23-04896-f005:**
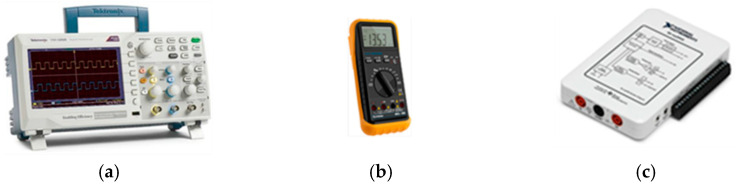
Measurement equipment: (**a**) oscilloscope; (**b**) multimeter; and (**c**) NI myDAQ.

**Figure 6 sensors-23-04896-f006:**
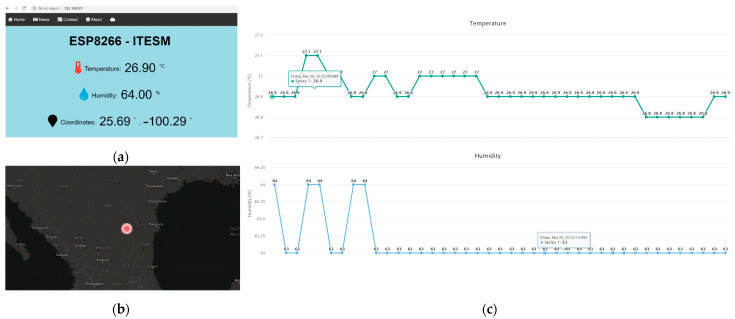
HTTP dashboard components: (**a**) displayed indicators; (**b**) real-time map; and (**c**) graphs of temperature and humidity.

**Figure 7 sensors-23-04896-f007:**
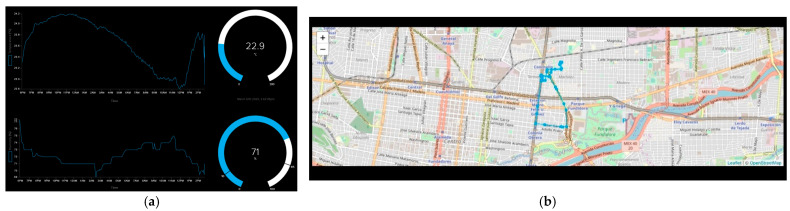
MQTT dashboard: (**a**) displayed indicators; (**b**) map with location and routes.

**Figure 8 sensors-23-04896-f008:**
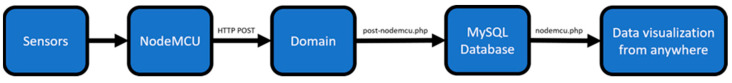
Database block diagram.

**Figure 9 sensors-23-04896-f009:**
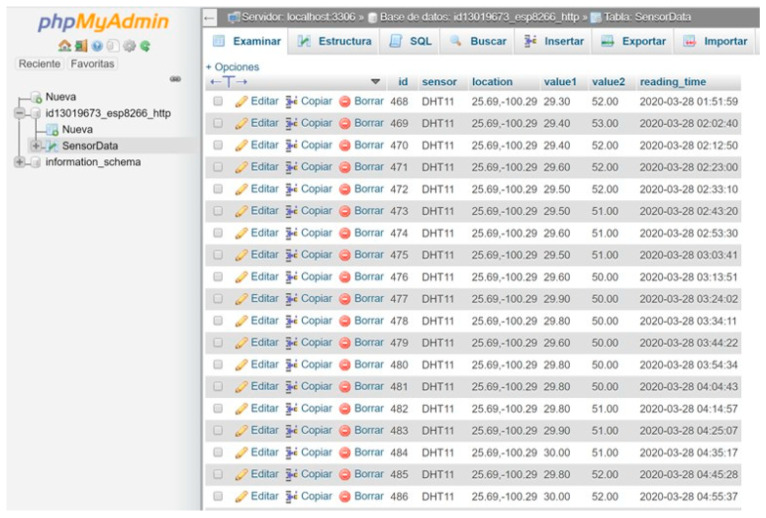
Database of recorded readings.

**Figure 10 sensors-23-04896-f010:**
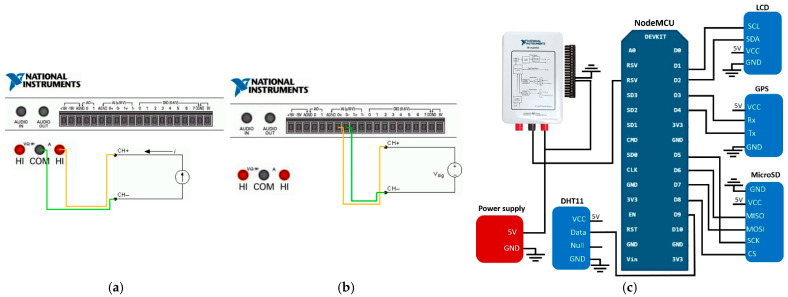
Schematics for the measurements with the NI myDAQ: (**a**) current measurement; (**b**) voltage measurement; and (**c**) complete schematic for the electronic platform.

**Figure 11 sensors-23-04896-f011:**
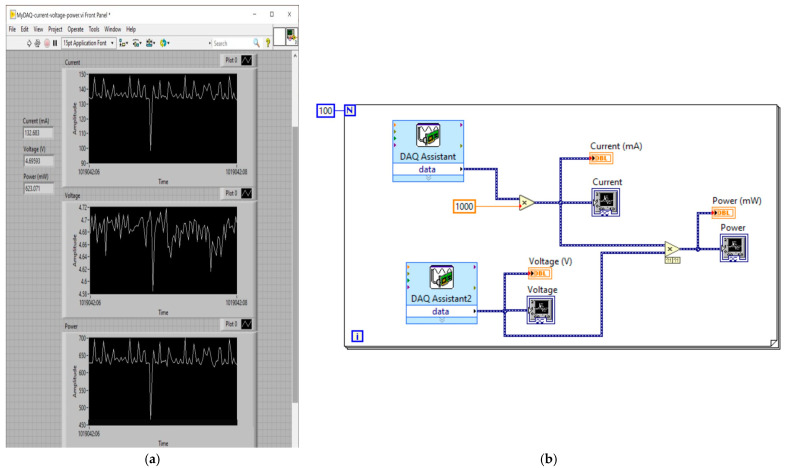
LabVIEW programming: (**a**) front panel; (**b**) block diagram.

**Figure 12 sensors-23-04896-f012:**
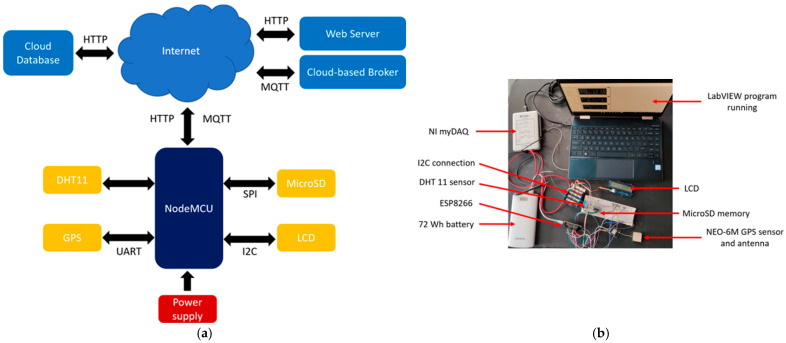
System prototype: (**a**) block diagram; (**b**) physical setup for experimentation.

**Figure 13 sensors-23-04896-f013:**
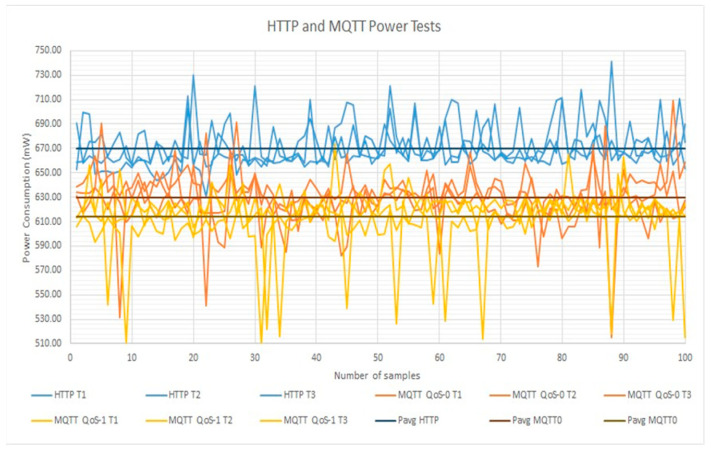
Graph of the power consumption in HTTP and MQTT protocols.

**Figure 14 sensors-23-04896-f014:**
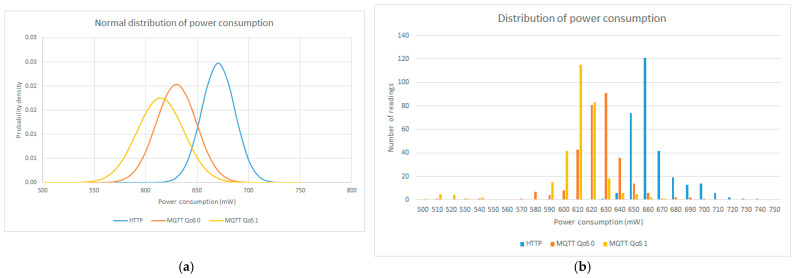
Power consumption in HTTP and MQTT: (**a**) normal distributions; (**b**) real distributions.

**Figure 15 sensors-23-04896-f015:**
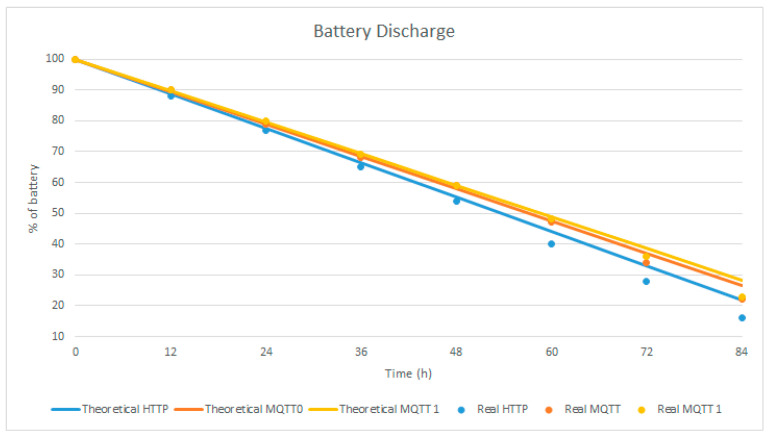
Battery life.

**Figure 16 sensors-23-04896-f016:**
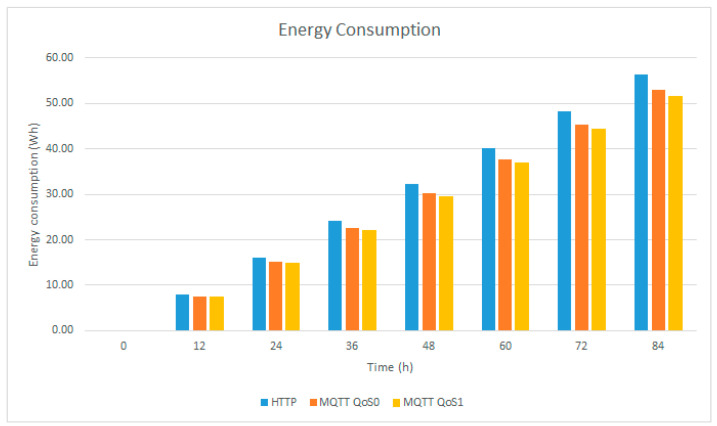
Energy consumption.

**Figure 17 sensors-23-04896-f017:**
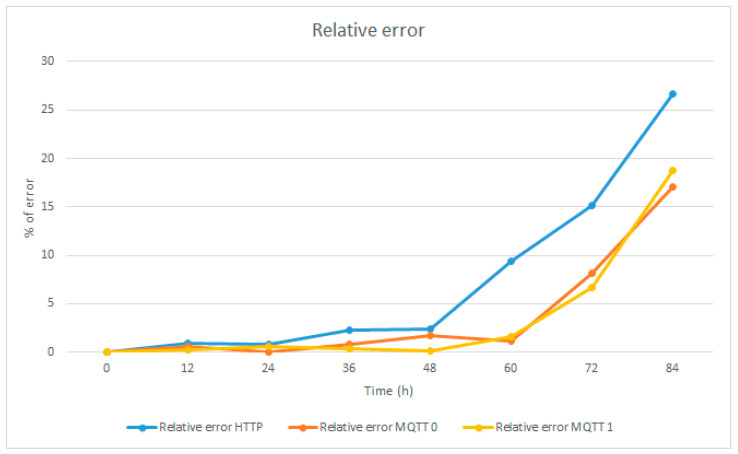
Relative error percentage.

**Table 1 sensors-23-04896-t001:** Components and price of the electronic platform.

#	Component	Unit Price (USD) ^1^	Price (USD) ^1^
1	NodeMCU	2.00	2.00
1	DHT11 temperature and humidity sensor	0.77	0.77
1	NEO 6M GPS sensor	3.50	3.50
1	MicroSD card adapter	3.25	3.25
1	4GB SanDisk microSD card	3.00	3.00
1	Grove LCD RGB backlight	9.09	9.09
1	Breadboard	1.00	1.00
		Total	22.61

^1^ The prices are expressed in dollars.

**Table 2 sensors-23-04896-t002:** Power consumption of HTTP and MQTT protocols.

Protocol	Description	Test	Imin	Imax	Iavg	Vmin	Vmax	Vavr	Pmin	Pmax	Pavg	σ	Energy
		(#)	(mA)			(V)			(mW)				(J)
HTTP	Only NodeMCU	1	36.25	83.73	78.00	4.79	4.93	4.87	177.40	408.49	379.91	39.96	56.99
		2	38.19	83.84	77.53	4.92	4.93	4.93	188.07	412.68	381.85	44.29	57.28
		3	38.65	81.90	78.22	4.86	4.92	4.90	188.85	401.69	383.09	34.10	57.46
		Avg.	37.70	83.16	77.91	4.85	4.93	4.90	184.77	407.62	381.62	39.45	57.24
	Complete system	1	136.45	154.02	139.29	4.75	4.88	4.80	654.88	741.21	668.89	14.98	100.33
		2	137.25	150.26	142.05	4.51	4.76	4.72	630.55	711.82	670.07	18.56	100.51
		3	136.79	152.88	140.20	4.74	4.86	4.79	664.59	664.73	671.52	15.04	100.73
		Avg.	136.83	152.39	140.51	4.67	4.83	4.77	650.01	705.92	670.16	16.19	100.52
MQTT QoS 0	Only NodeMCU	1	78.82	79.73	79.28	4.71	4.80	4.77	372.24	382.22	377.93	1.89	56.69
		2	78.71	80.53	79.35	4.55	4.93	4.78	359.37	391.76	379.58	6.66	56.94
		3	78.71	80.19	79.43	4.75	4.85	4.82	376.85	388.02	383.01	2.05	57.45
		Avg.	78.74	80.15	79.35	4.67	4.86	4.79	369.49	387.33	380.17	3.53	57.03
	Complete system	1	123.78	151.28	137.37	4.27	4.69	4.59	573.36	708.99	630.95	16.82	94.63
		2	136.11	150.83	139.42	4.24	4.64	4.53	583.48	691.75	631.81	19.84	94.78
		3	112.26	142.61	136.32	4.41	4.69	4.60	515.07	663.91	626.39	21.54	93.96
		Avg.	124.05	148.24	137.70	4.31	4.67	4.57	557.30	688.22	629.68	19.40	94.45
MQTT QoS 1	Only NodeMCU	1	42.87	81.90	77.97	4.63	4.88	4.79	203.62	395.42	373.71	34.43	56.06
		2	78.93	80.65	79.52	4.71	5.02	388.96	374.26	399.77	388.96	3.41	58.34
		3	79.16	80.53	79.46	4.83	5.03	4.88	383.79	399.03	387.57	2.60	58.14
		Avg.	66.99	81.03	78.99	4.72	4.98	132.88	320.56	398.07	383.41	13.48	57.51
	Complete system	1	113.28	148.32	135.71	4.47	4.58	4.55	513.58	673.47	616.80	23.87	92.52
		2	114.88	144.21	136.32	4.46	4.57	4.53	522.25	657.26	618.11	21.37	92.72
		3	113.28	143.41	135.28	4.41	4.59	4.50	509.55	649.26	608.08	21.55	91.21
		Avg.	113.82	145.31	135.77	4.44	4.58	4.52	515.12	660.00	614.33	22.26	92.15

**Table 3 sensors-23-04896-t003:** Expected battery life.

Protocol	Pavg (mW)	σ (mW)	Battery Life (h)	σ (h)	%/h
HTTP	670.16	± 16.19	107.44	± 2.59	0.9308
MQTT QoS 0	629.68	± 19.39	114.34	± 3.52	0.8746
MQTT QoS 1	614.33	± 22.25	117.20	± 4.24	0.8532
Only NodeMCU	381.73	± 18.81	188.61	± 9.29	0.5302

**Table 4 sensors-23-04896-t004:** Statistical parameters of the 300 samples per protocol.

Protocol	Min.	Max.	Average	σ	Variance	Median	Mode	Range
	P (mW)	P (mW)	P (mW)	P (mW)	P (mW)	P (mW)	P (mW)	P (mW)
HTTP	643.74	741.21	670.29	16.13	260.06	664.25	659.24	97.46
MQTT QoS 0	515.07	708.99	629.67	19.60	384.30	630.64	629.03	193.92
MQTT QoS 1	509.55	673.47	614.33	22.69	515.05	616.85	626.00	163.92

**Table 5 sensors-23-04896-t005:** Theoretical and real discharge of battery.

Hours	Theoretical (%)			Real (%)			Relative Error		
	HTTP	MQTT 0	MQTT 1	HTTP	MQTT 0	MQTT 1	HTTP	MQTT 0	MQTT 1
0	100	100	100	100	100	100	0	0	0
12	89	90	90	88	90	90	0.94	0.55	0.27
24	78	79	80	77	79	80	0.85	0.01	0.60
36	66	69	69	65	68	69	2.24	0.75	0.41
48	55	58	59	54	59	59	2.39	1.69	0.08
60	44	48	49	40	47	48	9.41	1.11	1.65
72	33	37	39	28	34	36	15.11	8.19	6.66
84	22	27	28	16	22	23	26.65	17.10	18.81

**Table 6 sensors-23-04896-t006:** Theoretical energy consumption.

Hours	Energy Consumption (Wh)		
	HTTP	MQTT QoS 0	MQTT QoS 1
0	00.00	00.00	00.00
12	08.04	07.56	07.37
24	16.08	15.11	14.74
36	24.13	22.67	22.12
48	32.17	30.22	29.49
60	40.21	37.78	36.86
72	48.25	45.34	44.23
84	56.29	52.89	51.60

## Data Availability

The data presented in this study is available in https://www.repositorionacionalcti.mx/recurso/oai:repositorio.tec.mx:11285/637503, accessed on 11 May 2023.

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
