# Peer review of "Comparative Analysis of Power Consumption between MQTT and HTTP Protocols in an IoT Platform Designed and Implemented for Remote Real-Time Monitoring of Long-Term Cold Chain Transport Operations"

_sensors, 2023, doi:10.3390/s23104896_

Round 1

Reviewer 1 Report

A design and validation of an electronic cost-efficient platform system for remote real-time monitoring is proposed in this paper.

Experimentation is done for HTTP and MQTT with different QoS levels to make a comparison and demonstrate the differences
in power consumption. The detailed design and the detailed comparison about the power consumption are well described.

As the authors point out "Although is well known that the MQTT consumes less power than HTTP, a comparative
analysis of power consumption with long-time tests and different conditions is not done yet" for this application. Therefore, I am not convinced that the detailed description of this specific application without new finding or new break-through is worthy publishing in a SCIE-indexed journal. This paper is more like a project report which elaborates on the detailed design and the performance evaluation.   It is more suitable in a conference.

Reviewer 2 Report

The paper presents a design and implementation of a cost-efficient IoT electronic platform for remote real-time monitoring of long-term cold chain transport operations using the NodeMCU module. 

The experimentation in the MQTT protocol with QoS 0 and 1 shows that the MQTT protocol saves more power than the HTTP protocol (6.03% and 8.33%, respectively) in long-time runs.

In addition, the paper also presents some strategies to optimize power consumption, opening up some new research directions related to the content presented.

The author should be more specific about the prediction model for the life cycle of batteries presented in the article. The authors also do not explain why they did not perform the experiments with QoS 2. If it's a future work, I'm eager to read the results.

The author of the paper needs to clarify the relationship between the research results and cold chain transport operations. The important term indicating the scope and application field of the paper "cold chain transport operations" is only mentioned twice in the whole paper, once in the Introduction and once in the Conclusion.

The paper is written in an easy to understand, well-organized manner. A few minor spelling and grammatical errors that need to be fixed.

Reviewer 3 Report

The authors design and validate an electronic cost-efficient platform system for remote real-time monitoring is proposed using the NodeMCU module, in which experimentation is done for HTTP and MQTT with different QoS levels to make a comparison and demonstrate the differences in power consumption.  This is a very nice and easy-to-read paper on a topic suitable for this journal. My main concerts are listed below:

1- The Introduction section has to be improved with a deeper analysis.

2-More technical papers about IoT are needed. 

[1] Development of an open sensorized platform in a smart agriculture context: A vineyard support system for monitoring mildew disease. Sustainable Computing: Informatics and Systems 28, 100309

[2] Microservice based scalable IoT architecture for device interoperability. Computer Standards & Interfaces, 84, 103697.

[3] A wot platform for supporting full-cycle iot solutions from edge to cloud infrastructures: A practical case. Sensors, 20(13), 3770.

[4] Internet of things. Manual of digital earth, 387-423.

Reviewer 4 Report

1.      In the abstract explaining using the proposed technique, please justify in the abstract regarding the method used.

2.      In the related works, the authors must focus on NodeMCU for IoT applications and Power consumption between HTTP and MQTT.The study need more studies related to MQTT and HTTP protocols for an IoT platform, add some latest related work .Please summarize them in table.

3.      Explain method section in order to clarify what are the methods that have used to achieve the research objectives and map them with research objectives.

  1. In the Experimental results, you are better to explain and analyze the data after listing it, not just explaining the concept.

5.      Figure 1, Figure 2 and Figure 3 are a bit blurry. Please consider replacing them with clear ones.

6.      The study need to add discussion about the results and benchmark it with other studies.

7.      Conclusion section needs to be added to the limitations of the study.

Try to use active voice in the article as much as possible.

Round 2

Reviewer 4 Report

No